# Mechanisms of Stem Cell Therapy in Spinal Cord Injuries

**DOI:** 10.3390/cells10102676

**Published:** 2021-10-06

**Authors:** Munehisa Shinozaki, Narihito Nagoshi, Masaya Nakamura, Hideyuki Okano

**Affiliations:** 1Department of Physiology, Keio University School of Medicine, 35 Shinanomachi, Shinjuku-ku, Tokyo 160-8582, Japan; shinozaki.mu@gmail.com; 2Department of Orthopedic Surgery, Keio University School of Medicine, 35 Shinanomachi, Shinjuku-ku, Tokyo 160-8582, Japan; nagoshi@keio.jp (N.N.); masa@keio.jp (M.N.)

**Keywords:** spinal cord injury, stem cell therapy, target of clinical trial, translational research, activities of daily living, complete injury

## Abstract

Every year, 0.93 million people worldwide suffer from spinal cord injury (SCI) with irretrievable sequelae. Rehabilitation, currently the only available treatment, does not restore damaged tissues; therefore, the functional recovery of patients remains limited. The pathophysiology of spinal cord injuries is heterogeneous, implying that potential therapeutic targets differ depending on the time of injury onset, the degree of injury, or the spinal level of injury. In recent years, despite a significant number of clinical trials based on various types of stem cells, these aspects of injury have not been effectively considered, resulting in difficult outcomes of trials. In a specialty such as cancerology, precision medicine based on a patient’s characteristics has brought indisputable therapeutic advances. The objective of the present review is to promote the development of precision medicine in the field of SCI. Here, we first describe the multifaceted pathophysiology of SCI, with the temporal changes after injury, the characteristics of the chronic phase, and the subtypes of complete injury. We then detail the appropriate targets and related mechanisms of the different types of stem cell therapy for each pathological condition. Finally, we highlight the great potential of stem cell therapy in cervical SCI.

## 1. Introduction

Spinal cord injury (SCI) affects 0.93 million (0.78–1.16 million) people worldwide each year [1], leaving irretrievable debilitating sequelae. Social and economic costs resulting from sequelae further justify the development of treatments for SCI. The central nervous system (CNS), i.e., the brain and spinal cord, is considered resistant to regeneration, and treatments to restore lost tissues and functions have not yet been established, with the exception of rehabilitation that can alleviate the condition of patients to a limited extent. Recent rehabilitation has been recommended to adapt to the pathophysiological heterogeneity of SCI, which depends on the time after injury, the degree of injury, and the spinal level of injury [2].

In recent years, stem cells, which have pluripotency and renewal ability and are rich in trophic factors, have been attracting attention as a source of effective treatment for various diseases, and clinical trials with many different types of stem cells have already been conducted for SCI [3,4,5,6]. In the present review, we categorize stem cell therapies that have been proposed for SCI and carefully refer to the pathophysiology of various spinal cord injuries. Next, we discuss the mechanism of action expected for each stem cell therapy, focusing on the following pathological conditions.

The first is complete chronic injury. Although this condition is considered the most difficult to treat, some recent studies have suggested that non-functional neural tissue remaining even in such so-called complete injury can provide some functional recovery once reactivated with appropriate treatment. Harkema and colleagues have reported that electrical stimulation could improve a completely injured state without replenishing the nerve tissue defect [7]. In addition, there have been several reports of spinal cord injuries in which no abnormalities were observed on MRI, although the injuries were entirely symptomatic [8,9,10]. One of the objectives of this review is to discuss the possibility of stem cell therapy in this specific subtype of complete chronic injury, which has not been discussed so far and remains unexplored.

Next, this review focuses on cervical spinal cord injuries. Cervical spinal cord injuries account for approximately 60% of spinal cord injuries, and since symptoms extend to the upper and lower limbs, the degree of sequelae is extremely severe. However, among the countless spinal cord therapeutic studies conducted in basic research, most of them target thoracic spinal cord injuries. Only a few studies have explicitly focused on cervical SCI, and even more surprisingly, very few studies have used the contusional injury model that is the most relevant for the clinical situation [11,12]. On the other hand, unlike thoracic and lumbar spinal cord injuries, the Activity of Daily Living (ADL) in cervical spinal cord injuries depends on the spinal level of injury: the more caudal the injury, the higher the ADL. Importantly, while the non-neural component of the lesion is crucial for the response to treatment, it is possible to obtain effective functional regeneration by regenerating the non-functional neural area in cervical SCI [5], even after so-called complete injury. Thus, we will discuss what can be expected from stem cell therapy for cervical spinal cord injuries.

## 2. Stem Cell Therapies for Spinal Cord Injury

The pathophysiology of SCI is complex, involving complex dynamic interactions between multiple cell types, and cannot be treated with unidimensional approaches, i.e., those that rely on a single mechanism of action. Due to their ability to duplicate, differentiate into multiple cell types, and secrete numerous trophic factors, stem cells are ideally suited to treat the multifaceted pathophysiology of SCI.

Stem cell therapies for SCI can be broadly categorized into two categories. The first category, which involves non-neural stem cells, includes bone marrow mesenchymal stem cells (BM-MSCs), umbilical cord MSCs (UC-MSCs), and adipose tissue-derived MSCs (AD-MSCs) [13,14,15,16,17,18,19,20,21,22]. Intravenous/intrathecal administration is often used, whereby engraftment to the injured area depends on a homing mechanism, and differentiation into neural cells is limited, and therefore the nervous replenishment of the lesion is also limited. However, neurotrophic factors secreted by MSCs have been shown to exert a therapeutic effect on the lesion [23,24]. This category of stem cell therapy with various types of MSCs, which we will refer to as supportive stem cell therapy in this review, constitutes the vast majority of clinical trials or studies in the field of SCI [18,19,20,25,26,27,28,29,30,31,32,33]. The fact that MSCs are usually easy to isolate for preparation compared with neural stem cells is a massive advantage in regulatory procedures.

The second category of stem cell therapies is based on stem cells that can generate neural cells. Olfactory ensheathing cells (OECs) [34,35,36,37,38,39,40,41,42,43,44], neural stem progenitor cells (NSPCs) [13,45,46,47,48,49,50,51,52,53,54,55,56,57], and neural progenitor cells (NPCs) derived from embryonic stem cells [58,59,60,61] or induced-pluripotent stem cells [62,63,64,65] are usually transplanted directly into the injured spinal cord. In that strategy, the transplantation procedure is invasive with special surgeries, and the preparation of the cells often requires a complex procedure. On the other hand, engraftment is the desired fate of the transplanted cells, replacing the lost neural cells and ultimately functionally substituting for them. This strategy will be referred to as loading therapy in this review.

## 3. Time-Dependent Pathological Changes after Spinal Cord Injury

The spinal cord is located in the spinal canal, which is surrounded by the spine. When the spine is deformed due to trauma, the spinal cord undergoes mechanical damage, which is called primary injury [66,67]. Both gray and white matter, with axonal rupture for the latter, are affected at the level of the lesioned spinal cord. Numerous white matter-descending tracts that control motor function can be impaired: the corticospinal tract, the rubrospinal tract, the reticulospinal tract, the tectospinal tract, the vestibulospinal tract, and monoaminergic spinal pathways. At the same time, the mechanical shock induces edema, bleeding and ischemia, the generation of hypoxia, the disruption of intracellular ion balance with abnormal sodium influx, and the disruption of the blood–spinal cord barrier due to blood vessel disruption [66,68,69,70]. This primary injury is an unavoidable consequence of the injury and cannot be targeted by treatment due to its suddenness.

Immediately afterward, a condition called secondary injury begins, resulting in further cell damage, nerve fiber rupture, and demyelination [71,72,73,74,75,76]. Secondary injury involves multiple cross-related events, such as the production of inflammatory cytokines, e.g., IL-1 and TNFα, by inflammatory cells, such as leukocytes and neutrophils, which infiltrate the lesioned area, the exacerbation of edema, and the development of neuronal glutamate excitotoxicity due to the disruption of ionic balance and membrane permeability [77]. Lipid peroxidation [78], the production of free radicals and nitrogen monoxide, the dysfunction of mitochondria and the electron transfer system due to oxidative stress, and excessive proteolytic activity of various proteases due to the increased concentration of intracellular calcium [79] also contribute to this secondary injury cascade. Furthermore, edema-induced swelling of the spinal cord within the spinal canal causes increased pressure and results in further ischemia and cell damage.

Previous rodent studies have indicated that oligodendrocytes (OLs) are damaged after SCI and are lost due to cell death, resulting in the demyelination of areas of white matter [80,81]. Experiments in rats have shown that OL death begins as soon as 15 min after spinal cord contusion and lasts for at least three weeks. Although the kinetics may differ, SCI experiments in mice have confirmed this phenomenon with a reduction in the number of OLs within the first 24 h and OL death that persists from three to seven days after injury. It is also reported that the apoptosis of OLs might occur as a consequence of long-term demyelination of spared axons in the late phase after SCI, because the loss of OLs was observed in regions far distant from the lesion [80]. OL death has also been observed in tissues of human patients [72,82,83], indicating that it is common to multiple types of SCI, in various species, including humans [80,84].

The secondary injury persists for several weeks after the injury. While the suggestion of steroid treatment was once of interest, no effective treatment could be confirmed in phase III of clinical trials. However, reducing secondary injury and consequently limiting inflammation-associated cell death remains an important therapeutic objective because the mitigating effect of both supportive and loading stem cell therapies is promising. During the subacute phase, i.e., the transition from the acute phase to the chronic phase, caspase-mediated apoptosis, further axonal demyelination, the containment of the lesion by the fibrotic and glial scars, and extracellular matrix remodeling are progressively achieved. It is also during this phase that a partial Wallerian degeneration takes place, which is retrograde cell death due to the so-called axon rupture. The pathophysiology of the chronic phase is also associated with limited natural regeneration and will be explained in the next section.

## 4. Factors Inhibiting Neuronal Regeneration

Axons do not regenerate spontaneously after SCI even after inflammation has subsided because the injured area is not filled naturally by tissues endowed with regenerative capacity. Astrocytes proliferate but are mainly recruited to the lesion site to make a glial border. Endogenous oligodendrocyte progenitor cells (OPCs) are reported to proliferate and mostly differentiate into oligodendrocytes [85,86], myelinating axons for several months. However, neurons have a small number of intrinsic precursors and do not compensate sufficiently for the injured part. The Wallerian degeneration of nerves whose axons are torn at the injured part is limited, and a considerable number of those cell bodies remain. Nevertheless, in addition to internal factors such as decreased axon outgrowth due to decreased intracellular cAMP concentration [87], external factors, including a lack of supportive neurotrophic factors, a cystic cavity in the center of injury that deprives axons of a physical substrate for their elongation, astrocytes, and inflammatory cells hinder axon regeneration [88,89,90,91]. In wounded tissues, scars are made up of various cells and extracellular matrix (ECM). In many organs, scar formation is directly associated with the recovery of tissue function. In contrast, the process of tissue scarring in the CNS is more ambivalent than in other tissues and does not unequivocally induce regeneration.

Specifically, scar formation in the injured CNS jointly involves fibroblast-like cell fibrosis, local inflammation by foamy macrophages that cannot process debris optimally, and corralling of the center of the lesion by reactive astrocytes [92,93]. The structure of the scar is then stabilized by various secreted ECM molecules, such as chondroitin sulfate proteoglycan (CSPG). In addition, the fragmentation of damaged axons caused by Wallerian degeneration generates debris, resulting in the extracellular deposition of myelin-related molecules (MAG, Nogo, OMgp). Together with CSPG, these molecules inhibit neuronal regeneration and neuroplasticity in the long term.

If functional tissue remains at the level of injury, the injury is classified as incomplete (Figure 1A). In natural recovery, the host axons sprout in the remaining tissue and create a detour through the surrounding healthy tissue [94,95,96,97,98,99,100,101]. After incomplete SCI in nonhuman primates, neurons of the corticospinal tract descending to the contralateral white matter have been shown to extend axons across the midline of the spinal cord and reshape the circuit. Similar phenomena have been reported for other descending tracts, such as the rubrospinal tract and the reticulospinal tract [102,103,104]. In those animal models, the number of axons in the descending tracts with residual axons is thought to recover partially due to lateral elongation after SCI. On the other hand, the total length of dendrites tends to increase after SCI, and this effect has been reported to be enhanced by glial cell-derived growth factor (GDNF) administration [105]. This change in the length of dendrites may be due to the elongation of individual dendrites, an increase in the number of dendrites, or a combination of both. Consistently, an increase in the size of the dendritic spines is observed concomitantly. Collectively, these changes suggest a highly plastic adaptation of the tissue around the injured area. In a hemisection model of SCI, spontaneous lateral axonal sprouting has been reported to occur around the injured site, and a new spinal cord circuit bypasses the site of the injury to generate a relay. In other words, in the case of incomplete injury, regeneration would certainly benefit from an environment that is more permissive to fiber extension in the residual tissue or reactive tissue. It seems reasonable to consider that both supportive and loading stem cell therapies can contribute to this objective.

When functional tissue does not remain in the injured area, the lesion is classified as a complete injury. Clinically, complete SCI is defined as the complete loss of motor and sensory function below the level of injury, with 40% of cases being A in the ASIA classification at the time of onset, but recovery is extremely limited even with conventional treatment and rehabilitation [106]. Hence, the proportion of patients with ASIA A is even higher in the chronic phase. However, recently, there have been reports of encouraging results following spinal cord epidural electrical stimulation or supportive stem cell therapy in complete SCI patients ASIA A [7]. How is function restored by a treatment method that does not supplement missing tissue in an injury where there is no residual functional tissue? Conversely, there are patients with severe symptoms whose spinal cord MRI reveals only a few abnormalities (SCIWORA) [8,9,10]. SCIWORA was originally a term used to describe spinal cord injury without bone abnormalities, but with the recent development of MRI, spinal cord injuries without abnormalities inside the spinal cord as well have also been reported as SCIWORA in a broader sense. Considering these observations, it is conceivable that there are two types of complete injuries: pseudo-complete SCI, in which a non-functional reactivable tissue remains, and true complete SCI, in which no reactivable tissue remains (Figure 1B,C) [107,108]. In pseudo-complete injury, functional recovery is expected by modifying the reactivable tissue with new therapies. Pseudo-complete injury is a promising target not only for loading stem cell therapy but also for supportive stem cell therapy.

## 5. Mitigating Neurotrophic Factors

Neurotrophic factors secreted from stem cells exert neuroprotective and anti-inflammatory effects [13,14,53,59,109]. In this regard, the optimal transplantation window is in the acute and subacute phases, when cell death and inflammation are maximal. Since the neuroprotective and anti-inflammatory effects start soon after transplantation, the improvement observed in motor function in experimental animals is usually observed rapidly after stem cell transplantation in the subacute phase.

Accordingly, immunostaining analyses have shown that transplanted neural stem cells reduce the accumulation of neutrophils and iNOS+/MAC-2+ activated macrophages in lesion areas [110]. RT-PCR analysis has also confirmed the downregulation of inflammatory cytokines such as TNF-α, IL-1β, IL-6, and IL-12 in this context. It is suggested that transplanted cells promote functional recovery after SCI by suppressing inflammation-induced secondary damage around the injured site.

Bone marrow mesenchymal stem cells have also been shown to improve function after SCI through mechanisms of neuroprotection via the secretion of neurotrophic factors, and the stabilization of the blood–spinal cord barrier [15,111]. RNA profiling was analyzed in the motor area of rats injected with BM-MSCs after SCI. The modification of the expression levels of potassium voltage-gated channel interacting protein 2, sodium voltage-gated channel beta subunit 3, and phosphodiesterase 10A indicates behavior-related changes that may be involved in the recovery of motor function after the systemic administration of BM-MSCs [112]. Therefore, the mitigating effect of the secretion of neurotrophic factors in supportive stem cell therapy and loading stem cell therapy appears to be a powerful mechanism of action, especially during the acute and subacute phases after SCI.

## 6. Modulating Neurotrophic Factors

In addition to suppressing inflammation and neuroprotection in the acute and subacute phases, neural stem cell-secreted neurotrophic factors have a continuous positive effect on neurons and astrocytes and result, for example, in the activation of the residual reactivable tissue remaining around the injury, which will function stably due to long-term potentiation. This is an effect that can be expected immediately after transplantation in both the subacute and chronic phases. Several trophic factors, such as brain-derived growth factor (BDNF), neurotrophin-3 (NT-3), nerve growth factor (NGF), fibroblast growth factor (FGF), and GDNF seem to exert a beneficial action on this mechanism [14,16,59,109]. NT-3 induces the outgrowth of axons in the corticospinal tract, while both NT-3 and NGF promote the outgrowth from the reticulum and red nucleus, respectively. BDNF induces lateral axon elongation in the rubrospinal tract, the reticular spinal tract, the vestibular spinal tract, and the anterior raphespinal tract. Using an experimental design that is not compatible with the clinical situation of SCI, a recent study in mice has identified three essential factors necessary for axon growth beyond the injured site. Osteopontin, insulin-like growth factor 1, and ciliary-derived neurotrophic factor (CNTF) were expressed in spinal neurons using a viral vector prior to injury. In addition, fibroblast growth factor-2 (FGF2) and epidermal growth factor were used to increase supportive substrates of axon growth such as laminin, and GDNF was released from synthetic hydrogel biomaterial depots placed caudally to chemoattract spinal neurons [113]. The results show that in such artificial conditions, a significant number of spinal axons could stretch across the center of injury.

In addition to anatomical regeneration, neurotrophic factors are also expected to promote the regeneration of functional circuits [114,115]. Synaptic activity is known to be modulated by neurotrophic factors [116,117,118], and the same action is anticipated for stem cell-secreted trophic factors. This action of neurotrophic factors on circuit activity is not limited to the residual circuits of the host. It will also likely increase the neural activity of new circuits formed between the host and transplanted cells.

An important question is related to the secondary effect of functional recovery of the spinal cord. After SCI, sprouting and rewiring induce the rewriting of the brain map, and functional substitutions between territories occur in the cerebral cortex [96,97,100,102]. Functional MRI studies have shown that changes over time are drastic [119]. In addition to the cerebral cortex, the circuits that descend from the cerebrum and brain stem to the spinal cord also change significantly after injury [120,121,122]. Therefore, if transplantation treatment is effective, these circuits are expected to change upon regeneration. For example, the motor-sensory area of the lower limb region, which is considered to disappear after a thoracic SCI, may reappear in other areas. In the case of dorsal injuries, the ventral reticular spinal tract tends to remain, so the cortico-reticular tract may become thicker in the hypothesis of an adaptative regeneration. Similarly, mild damage would also improve the cortico-red nucleus tract. The circuits that can cause such compensatory changes include the corticospinal tract, the rubrospinal tract, the reticulospinal tract, the vestibulospinal tract, the tectospinal tract, and the monoaminergic spinal tract [95,97,122], as well as those superior circuits such as the cortico-red nucleus tract, the cortico-reticular formation tract, and the cortico-raphe nucleus tract [123,124]. On the caudal side of the injury, it is expected that the sensory circuits that connect with the motor neurons will also be modified [96,98,122].

The long-term action of neurotrophic factors should not be underestimated. For example, sustained overexpression of NGF in the posterior horn results in the formation of inappropriate neural connections and leads to a severe nociceptive state in rats [125]. Therefore, in future preclinical studies, it is very important to define the appropriate combinations of proper growth factors and implement precise temporal administration methods to induce controlled axonal regeneration with minimal side effects.

## 7. Remyelination

Remyelination-promoting therapies are based on the premise that the function of residual demyelinated and dysfunctional nerve fibers can be restored by remyelination [126,127,128,129]. In the context of supportive stem cell therapy, this remyelination strategy requires that functional oligodendrocytes are still present. As a mechanism of action, remyelination is relatively slow because the formation of mature myelin sheath-wrapped axons is considered to take at least three weeks [130]. In the case of loading stem cell therapy, the necessary oligodendrocytes originate from the transplanted stem cells, which means that an additional period is necessary for the differentiation of the transplanted cells into mature oligodendrocytes.

A vast amount of data indicate that demyelination is the direct cause of functional symptomatic deterioration in patients with multiple sclerosis [131,132,133,134,135]. Experimentally, L-α-lysophosphatidylcholine (LPC)-induced demyelinating lesions in the thoracic spinal cord caused decreased myelin binding protein (MBP) staining, resulting in deteriorated motor function assessed by two types of behavioral tests, the Beam walking test and the Ladder walking test [136]. It has also been reported that the therapeutic effect for thoracic SCI in mice is reduced when the transplanted neural stem cells have been prepared from MBP-free Shiverer mice, demonstrating that remyelination is a key mechanism of the motor recovery in this model [130].

In other stem cell therapies, electron microscopy analyses indicate that more myelin is derived from direct transplantation of OPCs than from NSPCs [49,50,57,58,137]. Strategies to transplant modified OPCs have been reported. For example, Cao et al. infected OPCs with a vector expressing CNTF and transplanted the resulting cells nine days after injury to the thoracic spinal cord [50]. The survival rate of transplanted CNTF-OPCs increased four times compared to control OPCs. As expected, the transplanted OPCs reportedly formed a central myelin sheath around the axons of the injured spinal cord. Furthermore, in rats that received CNTF-OPCs, the response to transcranial magnetic stimulation and the recovery of motor function of the hindlimbs were significantly promoted.

However, in a recent review, the overall degree of functional deterioration caused by demyelination after SCI and to which extent this demyelination contributes to recovery were called the “remyelination enigma” [4]. For example, it has been reported that genetically modified mice incapable of remyelinating recovered normally after contusive SCI [138]. It has been argued that the roles of demyelination and remyelination should be supported by further studies.

## 8. Relay Mechanism

Regarding loading stem cell therapy, the so-called relay mechanism is the most promising mechanism but also the most difficult to implement [47,65,139,140]. For example, in the case of motor function, the axons of the upper neurons that survive and the neurons derived from transplanted cells must first form synapses. Then, these connected graft-neurons must extend the axons caudally and form synapses with anterior horn neurons and/or motor-related interneurons. Considering the complexity of such a regeneration scheme, it is not surprising that the complete proof of concept remains to be established.

As mentioned above, the brain map is significantly remodeled after SCI, as are the circuits from the brain to the brainstem [96,97,100,102,120,121,122]. In other words, there is a possibility that the original anatomical and functional circuits located rostrally to the injury disappear or change location. In addition, since the transplanted area is rich in trophic factors secreted by transplanted cells, axon invasion and synaptogenesis from the cranial host can be expected naturally, but the extension of the axon from the injured part to the caudal side requires a chemoattractant on the caudal side. It is reported that stem cells transplanted into the brain cannot extend their axons outward without an extrinsic chemoattractant [141], and the same is expected in the spinal cord. On the other hand, if enough fibers can extend caudally, a functional circuit would be formed with caudal host tissue [120]. During CNS development, a phenomenon known as pruning, i.e., the elimination of supernumerary nonactive connections, allows the selection of only the functional circuits [142]. This mechanism is expected to be involved in the regeneration processes that follow stem cell therapy.

To date, there are many examples of circuits formed between the host and transplanted cells [47,62,65]. The first line of evidence comes from the juxtaposition of neurites derived from transplanted cells labeled by β-III tubulin and hNu antibodies, and the presynaptic region of Bassoon-stained host neurons in the mouse spinal cord. Immunoelectron microscopy confirmed this interaction by visualizing transplanted cells through GFP expression and host mouse synapses. Furthermore, reports using the rabies virus as a transsynaptic tracer have shown that transplanted cells can form synapses in the brain stem and the caudal side. However, there are still no reports that directly show that upper neurons are bridged to lower neurons via transplanted cells.

The relay mechanism requires sufficient time for the axon to gradually elongate before any effect can be expected after the transplantation. The axon elongation rate of transplanted cells has been evaluated to ~1–2 mm/day, representing approximately 10 days to travel a distance of 20 mm from a thoracic injury to the lumbar spinal cord of a rat, not including the prior period for the differentiation of neural stem cells into neurons. How much time would be necessary for a human patient? The relay mechanism is the only mechanism that bridges the upper and lower circuits through true complete injury, but its practical realization is still a long way ahead and requires further research.

## 9. Specificity of Cervical Spinal Cord Injury

There are many reasons to explain the scarcity of clinically relevant reports on the treatment of cervical spinal cord injuries: the difficulty of assessing upper limb function, respiratory impairment, the spinal shock, and the complexity of managing quadriplegia, associated with ethical and care issues [3,143,144]. However, cervical spinal cord injuries account for 60% of spinal cord injuries [145] and represent a more severe clinical condition because they affect both upper and lower limbs. Among spinal cord injuries, the social need for treatment is the highest for cervical spinal cord injuries. Meanwhile, in the cervical cord, the innervated muscles of the upper limbs are relatively clearly separated for each medullary segment. Furthermore, since the upper limbs have more functions than the lower limbs, the ADL is completely different depending on the level of disability (Figure 2A) [146]. For example, if the functional range extends to C6, it will be possible to drive a car. Segmented functionalization of the cervical spinal cord is the most promising target because a significant therapeutic outcome, such as ADL in clinical trials, can be achieved by slight shifts in the medullary segments involved.

We have mentioned the existence of pseudo-complete injury in which some non-functional but reactivable tissue remains, even in functionally complete injury. Cervical spinal cord injuries include incomplete injuries, pseudo-complete injuries, and true complete injuries. In incomplete and pseudo-complete injuries, the mechanism of stem cell therapy is expected to involve mostly tissue-reactivating neurotrophic factors, as mentioned previously. For example, ASIA grade A patients in whom residual tissue was observed on MRI, i.e., corresponding to pseudo-complete injury, had a functional improvement of the injured caudal side by the administration of BM-MSCs [147]. When considering true complete injury of thoracic and lumbar spinal cord, the lesion is devoid of nerve tissue, and functional recovery with stem cell therapy would exclusively depend on the relay mechanism. In contrast, in a true complete injury of the cervical cord, the functional medullary level is expected to be widened by functionalizing reactivable tissue on the cranial side of the injury, which will directly lead to improved ADL (Figure 2B).

For the regenerative treatment of true complete injury, the usual objective is to restore the function of the caudal side of the injury. However, in the case of true complete injury of the cervical cord, the success of clinical trials should be evaluated with appropriate primary objectives, such as an expansion of the functional level on the cranial side of the injury and an accompanying increase in ADL.

## 10. Combination with Modified Rehabilitation

Multiple clinical trials have been conducted for SCI, but to date, only those related to rehabilitation had beneficial outcomes. There are few comparative clinical reports on combinational treatment with stem cell therapy because it is ethically difficult to treat SCI patients without rehabilitation. As an experimental study, it is reported that the transplantation of NSPCs and conventional treadmill rehabilitation with thoracic SCI mice in the chronic phase demonstrated the functional recovery of hindlimbs, suggesting the synergic effect at the lumbar enlargement [148].

Modified rehabilitations, which intend to modulate neural circuits, have been reported [149]. It is relatively easy to make an intervention on peripheral nerves, and wearable devices have been developed for functional electrical stimulation (FES) [150]. FES detects slight contractions of muscles and turns them into large stimuli, and repetitive cycles of inputs and outputs enhance the neural circuits. The Robot Suit HAL (hybrid assistive limb) expands this feedback using exoskeletons, and converts the electromyogram into a mechanical output to support joint movement and augment the neural circuit through feedback from sensory inputs [151]. Furthermore, as described above, neural plasticity exists not only around the lesion area, but also around the superior neural circuits, and there are studies describing the effectiveness of neurofeedback using electroencephalogram [149,152]. As the synergic effect was achieved with conventional rehabilitation, a combinational protocol of stem cell therapies with neurofeedback rehabilitation will be described in a future study.

## 11. Combination with Electrical/Magnetic Stimulation

Electrical/magnetic stimulation therapy as well as stem cell therapy has recently gained much attention [12,122,153,154,155,156,157,158,159,160,161,162,163,164,165,166,167,168]. Different types of stimulation have been applied to treat SCI: transcranial magnetic stimulation, therapeutic electrostimulation, peripheral nerve electrostimulation, transcutaneous nervous electrostimulation, subdural electrostimulation, epidural electrostimulation, subdural electrostimulation, deep-brain stimulation, and brain surface epidural electrostimulation that mimics transcranial magnetic stimulation.

In the case of methods that directly stimulate the spinal cord, such as epidural electrostimulation, subdural electrostimulation, and intraspinal electrostimulation, the stimulation of the input and output circuits of the spinal cord promotes the adjustment of posture and intensity of effort, and restores spontaneous motor control [169,170]. Complications of SCI such as flexor spasms, hyperreflexia, and spasticity may also result from inappropriate lateral sprouting and synaptic hyperplasia of motor neurons that have lost cortical input after injury. It is possible that electrostimulation sufficiently activates motor neurons and suppresses this excessive afferent synaptogenesis. Initially, these electrostimulation treatments were thought to modify and improve the remaining neural circuits. However, there are reports of effective electrostimulation in patients with ASIA A (complete injury) [7], suggesting that non-functional but reactivable tissue is also activated. There are reports describing the combinational effect of magnetic stimulation and stem cell therapy, and synergistic effects of those stimulations with stem cell therapy would be expected.

## 12. Conclusions

By targeting the non-functional reactivable tissue around the injured area, different regeneration mechanisms of stem cell therapy can be expected, even without direct transplantation into the injured area. In addition, since treatment for cervical SCI has high social needs and is likely to be effective according to its pathology, it is necessary to conduct research with appropriate objectives.

## Figures and Tables

**Figure 1 cells-10-02676-f001:**
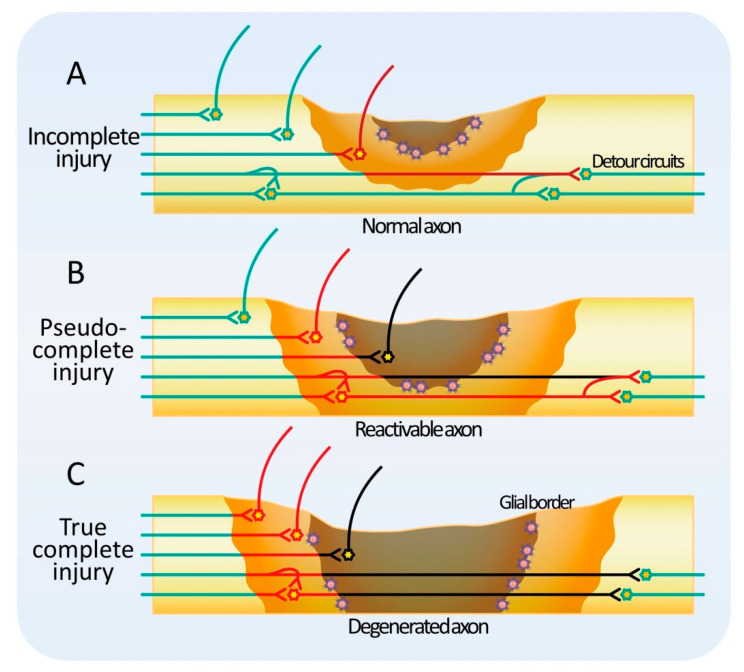
Classification of Spinal Cord Injuries. (**A**) Incomplete injury with residual functional tissue. Recovery is expected with conventional treatment. (**B**) Pseudo-complete injury. Non-functional but reactivable tissue (light brown) extends through the spinal cord, breaking the craniocaudal functional continuity. Recovery is difficult with conventional treatment. Functional recovery is expected by promoting axonal elongation and promoting neural activity in reactivable tissue. (**C**) True complete injury. Aneural tissue (gray) spans the entire spinal cord, and treatment of surrounding reactivable tissue at the rostral or caudal location does not provide a functional cranial connection.

**Figure 2 cells-10-02676-f002:**
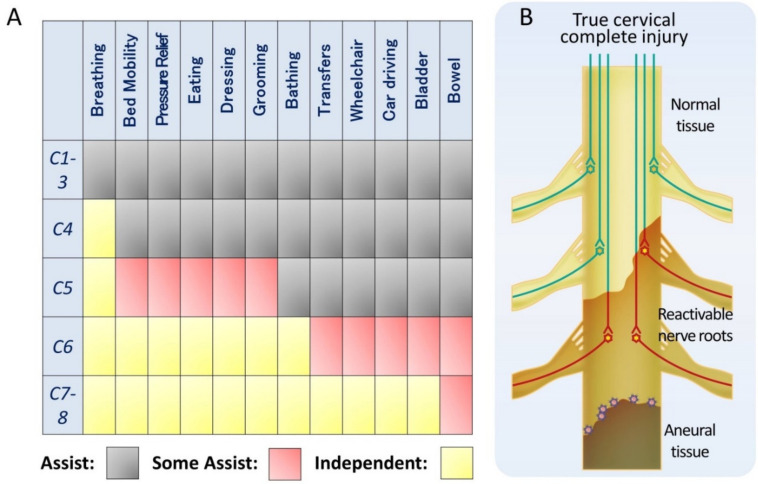
Specificity of Cervical Spinal Cord Injury. (**A**) Expected Activity of Daily Living (ADL) subcategories in various levels of cervical spinal cord injury. Unlike thoracic and lumbar injuries, slight differences in the spinal level of injury increase the independent ADL. Modified from reference [146]. (**B**) True complete cervical cord injury. The aneural area crosses the spinal cord (gray), but the reactivable tissue (light brown) around it remains. The ADL of patients is expected to increase as the reactivable tissue becomes functional from stem cell therapy.

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
