# Peer review of "Mechanisms of Stem Cell Therapy in Spinal Cord Injuries"

_cells, 2021, doi:10.3390/cells10102676_

Round 1

Reviewer 1 Report

The manuscript from Shinozaki et al. provides a nice overview about the mechanisms behind stem cell therapy for spinal cord injury repair, and how these mechanisms can be different, according to the specificity of the lesion.

The English is clear and the manuscript is well organized.

I believe this manuscript brings important aspects to the field and its publication is timely.

I would just call the attention to some details, namely:

Line 27: Include more recent epidemiological data in the paper from James et al. (Lancet Neurol. 2019 Jan;18(1):56-87.)

Line 33: Is there any reference for “modern rehabilitation”? Or is it just a term used to define modern-days programs of rehabilitation?

Line 75: Include more recent literature on the use of MSCs

Line 78: A few more studies could be mentioned or at least cited, regarding the use of MSC secreted factors for SCI repair (the so called “secretome”). Either from MSCs or any other type of stem cells.

Line 81: The same applies to clinical trials using MSCs, which I believe are scarcely cited throughout the manuscript.

Lines 101/102: I believe it is more correct to refer as “blood-spinal cord barrier”.

Line 133: Correct the sentence: “This is also during this phase…” probably by changing “This” to “It”.

Lines 142/143: Please revise the sentence: “…and a considerable number of cell bodies themselves remain.” It seems incomplete.

Line 170: Reference for GDNF administration example is missing.

Line 226: Could you please specify what do you mean by “motor field”?

Lines 284/286: References missing in this sentence: “as well as those superior circuits such as the cortico-rubrospinal tract, the cortico-reticular formation tract, and the cortico-anterior tract.”

Line 316: Reference for Cao et al. work is missing. It is reference no. 28 from the current version of the manuscript.

In order to increase the value of this manuscript I suggest the inclusion of the following papers/literature:

Asboth et al. Nat Neurosci. 2018 Apr;21(4):576-588.

More recent literature on OECs and NSCs (work from Mark Tuszynski for instance).

Reviewer 2 Report

This review seeks to describe the mechanisms by which cell therapy for spinal cord injury can be used to promote recovery, with a focus on ‘precision medicine’ – adapting the treatment to the type of injury. Overall, the review provides a good description of the pathophysiology of SCI, and mechanisms by which stem cell therapies could promote recovery. However, several problems with the manuscript somewhat reduce my enthusiasm including some language issues, a few missed key citations and limited utility given the staggering number of reviews on this topic area over the last 10-15 years. Despite this the authors do manage to make some novel insights into the treatment of complete and cervical spinal cord injury.

Current issues that need addressing:

  1. Language at times is a bit challenging. I recognize the authors are not from an English-first language speaking country and appreciate the effort they have clearly put in. However there are some issues (some examples outlined below) that need addressed and I recommend the authors thoroughly examine the text or get an English language first person to edit it.

Line 137 – ‘Factors Inhibiting The Natural Regeneration’

I don’t know what this means. I think this could be reworded to Factors Inhibiting Neuronal Regeneration

Line 210 – ‘Soothing Neurotrophic Factors’

Soothing means ‘having a gently calming effect’ – I don’t think this is what the offers intended here.

I think a better way to reword that would be Secretion of Beneficial Neurotrophic factors or something similar.

Use of the words ‘cure’ and ‘heal’ (lines 29, 46)

The word cure means ‘recovery or relief from a disease.’ This seems like an overstatement of where the current treatments (or lack thereof) for SCI are and should be reworded.

  1. Specific claims lacking citation:

This occurs somewhat frequently in the manuscript, for example in line 256-257 ‘Synaptic activity is known to be modulated by neurotrophic factors’ – this is a highly specific claim that lacks citations.

Additionally lines 267-268: ‘Since blood vessels are more vulnerable to ischemia than other CNS tissues’

I encourage the authors to examine the text and potentially find other examples of this.

The remyelination section lacks a critical citation that should be discussed and sheds further light on the ‘remyelination enigma.’ Genetically modified mice incapable of remyelinating recover normally following contusive SCI (Duncan et al. 2018) – doubting the role of remyelination in recovery from those injuries. This manuscript should be discussed and the implications for stem cell therapy to promote remyelination.

  1. The authors bring up the intriguing point at the very end that stem cell therapies may synergize with rehabilitation therapies to promote recovery. This should be expanded into an additional section – what are the mechanisms by which stem cell therapies could help rehabilitation to better drive recovery? Are their rodent or human studies which support this approach?

Minor Issues.

Line 121- Does the prolonged OL death result from loss of the underlying axons/neurons or is it truly a result of injury to the oligodendrocyte? Several studies have found prolonged oligodendrocyte death is most common in severed tracts where the axons are degenerating and is likely a consequence of this (eg Crowe 1997). This should be discussed.

Line 139-140: ‘The injured area is not filled naturally be tissue endowed with regenerative capacity’. The authors then go on to discuss how oligodendrocytes, astrocytes and neurons have limited number of intrinsic progenitors.

This is simply not entirely true. OPCs constitute roughly ~5% of the CNS and are capable of making enormous numbers of new oligodendrocytes (and Schwann Cells) after SCI. This should be mentioned.

Line 263-264: Is the increased expression of VEGF a good thing? Many tumours use VEGF to expand and stem cell therapies have been shown to be tumourigenic in some cases before. The authors should comment on whether a tumour was observed in this case.
